# Pollen and Floral Organ Morphology of 18 Oil-Tea Genotypes and Its Systematic Significance

Qian Yin [1] , Zhongfei Pan [1], Yanming Li [1], Huan Xiong [1,*], Joseph Masabni [2], Deyi Yuan [1] and Feng Zou [1,*]

1    Hunan Key Laboratory of Colleges and Universities of Oil Tea Breeding, Central South University of Forestry and Technology, Changsha 410004, China; 20221100088@csuft.edu.cn (Q.Y.); 20221100089@csuft.edu.cn (Z.P.); 20210100002@csuft.edu.cn (Y.L.); t20061123@csuft.edu.cn (D.Y.)
2    Texas A&M Agricultrue Life Research and Extension Center, 17360 Coit Rd., Dallas, TX 75252, USA; joe.masabni@ag.tamu.edu
*    Correspondence: t20202524@csuft.edu.cn (H.X.); t20142217@csuft.edu.cn (F.Z.)

**Abstract:** Oil-tea belongs to the *Camellia* genus, an important oil crop in China. However, oil-tea is taxonomically challenging due to its morphological variation, polyploidy, and interspecific hybridization. Therefore, the present study aimed to investigate the flower organs' morphology and pollen micro-morphology of 18 oil-tea genotypes in detail and discussed their significance for oil-tea taxonomy. The quantitative parameters of flowers were measured using Vernier caliper measurements. Pollen morphology was observed and photographed using scanning electron microscopy (SEM). The results indicated that the flower size varied significantly among the tested oil-tea genotypes, with the corolla diameter ranging from 42.25 μm in *C. meiocarpa* 'LP' to 89.51 μm in *C. oleifera* 'ASX09'. The pollen grains of oil-tea are monads and medium grade in pollen size. There were two types of polar views, including triangular or subcircular, with a polar axis length (P) ranging from 27.5 μm in *C. oleifera* 'CY67' to 59.04 μm in *C. mairei* (H. Lév.) Melch. var. lapidea (Y.C. Wu) Sealy. The equatorial views exhibited oblate, spherical, or oblong shapes, with an equatorial axis length (E) of 21.32 to 41.62 μm. The pollen exine sculpture was perforate, verrucate, and reticulate. The perforation lumina diameter (D) ranged from 0.29 μm in *C. magniflora* Chang to 1.22 μm in *C. yuhsienensis* Hu, and the perforation width (W) varied from 0.77 μm in *C. osmantha* to 1.40 μm in *C. gauchowensis* 'HM349', respectively. Qualitative clustering analysis (Q-type cluster) and principal component analysis (PCA) were conducted using eleven indexes of flower and pollen morphology, and the 18 oil-tea genotypes were classified into three categories. In addition, the correlation analysis showed that there was a significant correlation between pollen size and flower morphology or pollen exine sculpture. These results offer valuable information on the classification and identification of the 18 oil-tea germplasm resources.

**Keywords:** oil-tea; floral organ morphology; pollen morphology; scanning electron microscope

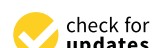



## 1. Introduction

Oil-tea (*Camellia oleifera*), a small evergreen tree of the *Camellia* genus, has a high oil content in its seed [1,2]. It is mainly distributed in southern China, with a planting area of over 4,000,000 hectares. Oil-tea is rich in unsaturated fatty acids, such as oleic acid and linoleic acid, making it nutritionally comparable to olive oil [3–6]. This advantage makes it a high-quality and healthy option as an edible vegetable oil. In China, some of the main oil-tea plants include *C. oleifera*, *C. oleifera* var. monosperma, *C. vietnamensis*, *C. yuhsienensis*, and others [2,7]. However, their diverse morphologies and a lack of interest in variety classification by farmers during planting pose a significant challenge to the development of the oil-tea industry [8,9].

Pollen morphological characteristics have always played an important role in plant taxonomy and variety classification [10–12]. Pollen features, which include the shape,

size, polarity, surface decoration, number, and type of germinating pollen organs, are less affected by environmental factors [13,14]. Many plant species have been classified according to their pollen characteristics, such as *Aletris* [12], *Veratrum* L. (Melanthiaceae) [15], *Vitaceae* [16], and *Gossypium* [17]. In *Camellia*, previous work also used pollen morphology to describe the relationships among taxa. For example, Wei et al. [18] observed the pollen morphology of 20 *Camellia* varieties and divided them into three categories according to their pollen features. According to the exine ornament and shape of pollen, Wang et al. [19] classified ten *Camellia* varieties into three groups. Xie et al. [20] found that *C. meiocarpa* had a close genetic relationship with the sect. *Camellia oleifera* Abel. and sect. *Camellia* (L.) by observing pollen morphology. Yuan et al. [21] described differences between pseudopollen and normal pollen by using the pollen morphological structure. Thus, these studies indicated that pollen characteristics are a powerful tool in identifying plants in the *Camellia* genus.

Although descriptions of anatomical characteristics and relational analyses of pollen in *Camellia* do exist, differences in these traits among the various oil-tea genotypes are still unclear. Therefore, in this study, we examined the floral organ and pollen morphology of 18 oil-tea genotypes by microscopy. Additionally, qualitative clustering analysis (Q-type cluster) and principal component analysis (PCA) of the pollen morphological characteristics were conducted. These findings provide detailed knowledge for the taxonomic and systematic identification of the *Camellia* genus.

## 2. Materials and Methods

### 2.1. Plant Materials

Eighteen oil-tea specimens were collected from the experimental site of Central South University of Forestry and Technology, located in Changsha, Hunan Province, China. These pollen grains were harvested from mature stamens during the full flowering stage, which occurred from November 2022 to February 2023. A total of twenty flowers were selected from each genotypic variety of oil-tea (Table 1; Figure 1). These samples of ploidy were described by Ye et al. [22] and Li et al. [23].

**Table 1.** Characteristics of flowering time and ploidy of selected oil-tea genotypes.

| Code | Genotype | Sampling Date | Ploidy | Code | Genotype | Sampling Date | Ploidy |
|------|----------|---------------|--------|------|----------|---------------|--------|
| no. 1 | *C. gauchowensis* 'HM19' | 2 December | 8n = 120x | no. 10 | *C. hainanica* | 6 December | 10n = 150x |
| no. 2 | *C. gauchowensis* 'HM349' | 25 November | 8n = 120x | no. 11 | *C. magniflora* Chang | 6 February | 8n = 120x |
| no. 3 | *C. oleifera* 'ASX09' | 2 December | 6n = 90x | no. 12 | *C. mairei* (H. Lév.) Melch. var. lapidea (Y.C. Wu) Sealy | 28 February | 2n = 30x |
| no. 4 | *C. hainanica* 'PX-6' | 26 November | 10n = 150x | no. 13 | *C. oleifera* 'HJ' | 6 December | 6n = 90x |
| no. 5 | *C. meiocarpa* 'ZX0907' | 2 December | 4n = 60x | no. 14 | *C. oleifera* 'HS' | 24 November | 6n = 90x |
| no. 6 | *C. oleifera* 'DY2' | 28 November | 6n = 90x | no. 15 | *C. oleifera* 'HX' | 26 November | 6n = 90x |
| no. 7 | *C. oleifera* 'DZ1H' | 15 December | 6n = 90x | no. 16 | *C. osmantha* | 22 November | 6n = 90x |
| no. 8 | *C. oleifera* 'CY67' | 22 November | 6n = 90x | no. 17 | *C. meiocarpa* 'LP' | 3 December | 4n = 60x |
| no. 9 | *C. gauchowensis* 'XW' | 15 December | 10n = 150x | no. 18 | *C. yuhsienensis* Hu | 24 February | 6n = 90x |

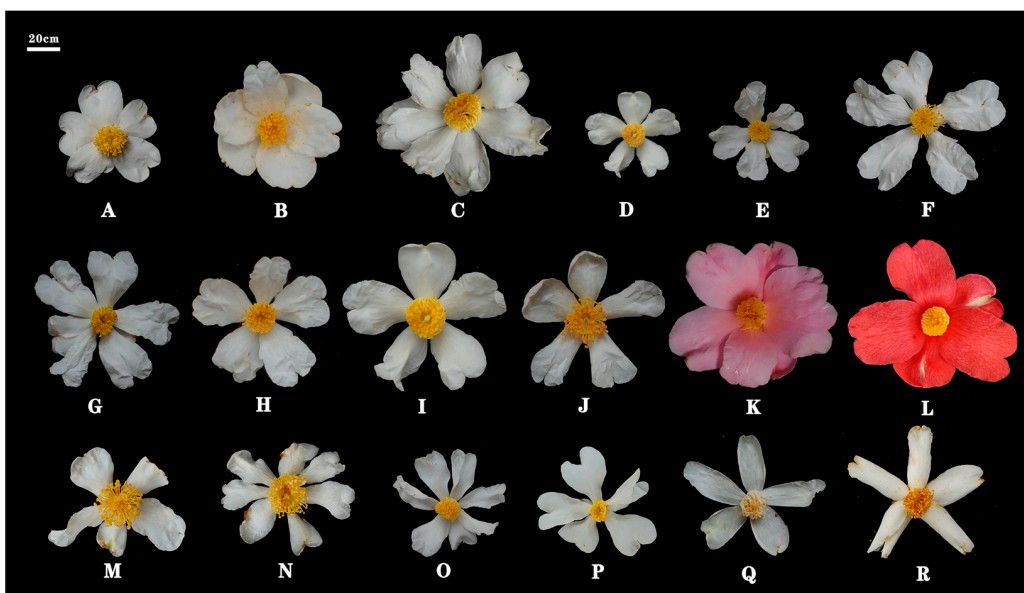

**Figure 1.** Flower morphology of 18 specimens of oil-tea *Camellia*. (**A**) *C. gauchowensis* 'HM19', (**B**) *C. gauchowensis* 'HM349', (**C**) *C. oleifera* 'ASX09', (**D**) *C. hainanica* 'PX-6', (**E**) *C. meiocarpa* 'ZX0907', (**F**) *C. oleifera* 'DY2', (**G**) *C. oleifera* 'DZ1H', (**H**) *C. oleifera* 'CY67', (**I**) *C. gauchowensis* 'XW', (**J**) *C. hainanica*, (**K**) *C. magniflora* Chang, (**L**) *C. mairei* (H. Lév.) Melch. var. lapidea (Y.C. Wu) Sealy, (**M**) *C. oleifera* 'HJ', (**N**) *C. oleifera* 'HS', (**O**) *C. oleifera* 'HX', (**P**) *C. osmantha*, (**Q**) *C. meiocarpa* 'LP', and (**R**) *C. yuhsienensis* Hu.

*2.2. Morphology Studies*

Ten morphological characteristics of the flower organ were measured from twenty samples of each oil-tea genotype. These included the petal color, corolla diameter (CD), anther number (AN), stamen cluster (SH), petal count (PC), style height (STH), pistil height (PH), stigma cracking, pistil, and stamen relative position.

The pollen grains were mounted on metallic stubs coated with gold-palladium and observed under a scanning electron microscope (JSM-6389LV, JEOL, Tokyo, Japan) [20]. Fifty samples of pollen from each genotype were randomly selected for monosomic photography. The characteristics of the pollen morphology were studied, including the numeric parameters and shape of pollen, presence of muri, polar outline, equatorial axis length (E), polar axis length (P), lumina diameter (D), muri width (W), germination furrow width (WG), the ratio of polar axis length to equatorial axis length (P/E), and the lumina diameter to muri width (D/W). The various pollen types were classified according to the criteria of the pollen shape types proposed by Halbritter et al. [24]. The classification criteria for the pollen shape depended on the shape type and the ratio of the polar axis to an equatorial axis (P/E). If P/E is >2, the pollen is classified as super subprolate. For P/E $1.14 \leq P/E < 2$, $0.88 \leq P/E < 1.14$, $0.5 \leq P/E < 0.88$, and $P/E < 0.5$, the pollen shape type was defined as prolate ellipsoid, sub-spheroid, oblate spheroid, and super oblate spheroid [9,24], respectively.

*2.3. Data Analysis*

Pollen parameter measurements were conducted using Image-Pro Plus 6.0 (Media Cybernetics, Silver Spring, MD, USA). The quantitative measurements were analyzed using SPSS 16.0 software to calculate the minimum, mean, standard error, and maximum of each sample character. The interrelationships among the eleven evaluated traits, including CD, AN, SH, PH, P, E, D, W, a (arc width of exine), b (arc height of exine), and WG, were performed using qualitative clustering analysis (Q-type cluster) and principal component analysis (PCA) by the SPSS v26.0. (IBM, Chicago, USA) software [18,25]. Pearson correlative analysis was used to test the significance between the floral organs and pollen morphology [17].

## 3. Results

### 3.1. Characteristics of Flower Organ Morphology

The floral morphology of the 18 oil-tea specimens is presented in Tables 2 and 3. The petal color of the 16 oil-tea genotypes was white, while that of genotypes no. 11 and no. 12 were pink and rose bengal, respectively. The corolla diameter (CD) ranged from 42.25 mm for no. 17 to 89.51 mm for no. 3. Among the tested samples, no. 3 had the longest corolla diameter, while no. 17 had the shortest. The number of petals ranged from 5 to 8. The corolla shape of no. 11 and no. 12 was campanulate, while the corolla of all other genotypes was rotated (Figure 1). Among these genotypes, no. 12, no. 10, and no. 11 had the highest stamen cluster height (SH) and anther number, while no. 18 had the lowest. Table 3 indicates that the pistil height (PH) of no. 12 was the highest among the 18 genotypes, while no. 18 was the shortest. The stigma of no. 2, no. 5, and no. 12 had 3 shallow cracks, while all the other genotypes had 3–4 shallow or deep cracks. In this study, we observed that the androecium of no. 6, no. 7, no. 8, no. 14, no. 15, no. 17, and no. 18 was found to be significantly higher than the pistil, while no. 1 and no. 16 had stamens shorter than the height of the pistil; however, all the other genotypes had similar heights between the stamen and pistil (Table 3).

**Table 2.** Flower morphology indices (mean ± SD) (n = 20) and description of floral organ characteristics of studied oil-tea genotypes.

| Code | Genotype | Petal Color | Corolla Diameter (mm) (CD) | Anther Number (AN) | Stamen Height (mm) (AH) | Petal Count (Petals per Flower) (PC) |
|---|---|---|---|---|---|---|
| no. 1 | *C. gauchowensis* 'HM19' | white | 59.11 ± 1.03 i | 112.75 ± 2.2 c | 16.10 ± 0.2 hij | 5 |
| no. 2 | *C. gauchowensis* 'HM349' | white | 69.54 ± 1.39 ef | 92.90 ± 0.96 def | 15.77 ± 0.21 k | 5 |
| no. 3 | *C. oleifera* 'ASX09' | white | 89.51 ± 2.11 a | 99.65 ± 4.03 d | 18.04 ± 0.25 ef | 6 (or 6 ± 1) |
| no. 4 | *C. hainanica* 'PX-6' | white | 66.43 ± 1.65 fgh | 81.50 ± 3.01 g | 15.51 ± 0.28 j | 6 (or 6 ± 1) |
| no. 5 | *C. meiocarpa* 'ZX0907' | white | 49.86 ± 0.73 j | 85.45 ± 3.06 efg | 17.07 ± 0.2 fgh | 6 (or 6 ± 1) |
| no. 6 | *C. oleifera* 'DY2' | white | 79.76 ± 1.67 bc | 91.90 ± 1.35 def | 17.77 ± 0.28 efg | 6 (or 6 ± 1) |
| no. 7 | *C. oleifera* 'DZ1H' | white | 84.15 ± 2.46 ab | 101.30 ± 1.3 d | 19.09 ± 0.33 cd | 6 (or 6 ± 1) |
| no. 8 | *C. oleifera* 'CY67' | white | 73.80 ± 1.19 de | 94.50 ± 2.8 de | 16.37 ± 0.43 hij | 5 (or 6) |
| no. 9 | *C. gauchowensis* 'XW' | white | 86.59 ± 1.67 a | 89.25 ± 4.15 efg | 15.52 ± 0.26 ij | 5 (or 6) |
| no. 10 | *C. hainanica* | white | 75.71 ± 1.22 cd | 168.50 ± 3.82 a | 19.81 ± 0.26 c | 6 (or 6 ± 1) |
| no. 11 | *C. magniflora* Chang | pink | 48.74 ± 1.73 jk | 150.75 ± 3.68 b | 36.54 ± 0.33 b | 6 (or 6 ± 1) |
| no. 12 | *C. mairei* (H. Lév.) Melch. var. lapidea (Y.C. Wu) Sealy | rose bengal | 47.88 ± 1.72 jk | 175.65 ± 2.4 a | 39.88 ± 0.45 a | 6 (or 6 ± 1) |
| no. 13 | *C. oleifera* 'HJ' | white | 64.70 ± 1.33 efgh | 70.90 ± 1.57 h | 12.00 ± 0.23 l | 5 |
| no. 14 | *C. oleifera* 'HS' | white | 66.88 ± 1.36 fg | 113.50 ± 2.93 c | 18.72 ± 0.26 de | 6 (or 6 ± 1) |
| no. 15 | *C. oleifera* 'HX' | white | 60.73 ± 2.19 hi | 115.60 ± 3.91 c | 16.94 ± 0.32 ghi | 6 (or 6 ± 1) |
| no. 16 | *C. osmantha* | white | 70.24 ± 1.93 def | 83.45 ± 1.54 g | 15.77 ± 0.37 j | 6 (or 6 ± 1) |
| no. 17 | *C. meiocarpa* 'LP' | white | 42.25 ± 1.37 k | 64.05 ± 3.02 h | 11.68 ± 0.12 l | 5 (or 6) |
| no. 18 | *C. yuhsienensis* Hu | white | 61.44 ± 1.58 ghi | 46.50 ± 1.07 i | 11.11 ± 0.23 l | 5 (or 6) |

Different letters within a column indicate significant differences at $p \leq 0.05$ based on Duncan's multiple range test. Abbreviations: CD = corolla diameter; AN = anther number; SH = stamen height; PC = petal count.

**Table 3.** Flower morphology indices (mean ± SD) (n = 20) and description of floral organ characteristics of selected oil-tea genotypes.

| Code | Genotype | Pistil Height (mm) (PH) | Style Height (mm) (STH) | Cracking state of Stigma and style | Pistil and Stamen Relative Position |
|---|---|---|---|---|---|
| no. 1 | *C. gauchowensis* 'HM19' | 13.49 ± 0.15 e | 9.84 ± 0.09 f | 4 deep cracks | A < G |
| no. 2 | *C. gauchowensis* 'HM349' | 13.95 ± 0.11 e | 11.24 ± 0.1 e | 3 shallow cracks | A > G (or A ≈ G) |
| no. 3 | *C. oleifera* 'ASX09' | 12.10 ± 0.42 f | 8.49 ± 0.42 gh | 4 deep cracks | A > G |
| no. 4 | *C. hainanica* 'PX-6' | 16.20 ± 0.2 cd | 12.55 ± 0.22 c | 3 deep cracks | A > G (or A ≈ G) |

**Table 3.** *Cont.*

| Code | Genotype | Pistil Height (mm) (PH) | Style Height (mm) (STH) | Cracking state of Stigma and style | Pistil and Stamen Relative Position |
|------|----------|-------------------------|-------------------------|-----------------------------------|-------------------------------------|
| no. 5 | *C. meiocarpa* 'ZX0907' | 12.39 ± 0.2 f | 9.47 ± 0.22 fg | 3 shallow cracks | A > G |
| no. 6 | *C. oleifera* 'DY2' | 10.90 ± 0.12 g | 7.75 ± 0.12 hi | 3 (or 4) deep cracks | A > G (or A ≈ G) |
| no. 7 | *C. oleifera* 'DZ1H' | 16.29 ± 0.18 c | 12.64 ± 0.19 cd | 5 (or 6) deep cracks | A > G |
| no. 8 | *C. oleifera* 'CY67' | 15.86 ± 0.31 cd | 12.32 ± 0.31 cd | 3 (or 4) deep cracks | A > G (or A ≈ G) |
| no. 9 | *C. gauchowensis* 'XW' | 15.27 ± 0.23 d | 11.67 ± 0.29 e | 5 (or 6) deep cracks | A > G |
| no. 10 | *C. hainanica* | 17.39 ± 0.14 c | 13.57 ± 0.13 c | 3 (or 4) deep cracks | A > G (or A ≈ G) |
| no. 11 | *C. magniflora* Chang | 31.03 ± 0.36 b | 27.59 ± 0.39 b | 3 (or 4) shallow cracks | A > G |
| no. 12 | *C. mairei* (H. Lév.) Melch. var. lapidea (Y.C. Wu) Sealy | 39.52 ± 0.49 a | 36.04 ± 0.51 a | 3 shallow cracks | A > G |
| no. 13 | *C. oleifera* 'HJ' | 13.62 ± 0.26 e | 9.95 ± 0.25 f | 3 (or 4) deep cracks | A > G |
| no. 14 | *C. oleifera* 'HS' | 16.12 ± 0.17 cd | 11.64 ± 0.54 de | 4 (or 5) deep cracks | A > G |
| no. 15 | *C. oleifera* 'HX' | 13.41 ± 0.38 e | 9.93 ± 0.31 f | 3 (or 4) deep cracks | A > G (or A ≈ G) |
| no. 16 | *C. osmantha* | 10.61 ± 0.09 g | 7.33 ± 0.11 i | 3 (or 4) deep cracks | A > G |
| no. 17 | *C. meiocarpa* 'LP' | 10.12 ± 0.11 g | 7.29 ± 0.07 i | 3 deep cracks | A > G (or A ≈ G) |
| no. 18 | *C. yuhsienensis* Hu | 4.83 ± 0.06 h | 2.93 ± 0.05 j | 3 deep cracks | A > G |

Different letters within a column indicate significant differences at $p \leq 0.05$ via Duncan's multiple range test. Abbreviations: STH = style height; PH = pistil height; A < G (or A ≈ G) = mostly androecium < gynoecium, with a few androecium ≈ gynoecium; A > G (or A ≈ G) = mostly androecium > gynoecium, with a few androecium ≈ gynoecium; A > G = androecium > gynoecium; deep crack = Stigma appears crack and the position of the crack reaches the position more than 1/3 style length; shallow crack = Stigma appears crack, which the crack reaches the position less than 1/3 style length.

### 3.2. Characteristics of Pollen Morphology

The pollen morphology of 18 oil-tea genotypes is presented in Figures 2–4, Tables 4–7. We observed that all genotypes displayed a triangular or subcircular shape in the polar view, while the equatorial view presented an oblate, oblong, and spheroid shape based on the P/E values (Table 5; Figures 2–4). The pollen grains exhibited symmetrical or radially symmetrical characteristics on both sides (Figures 2–4). The polar axis length (P) ranged from 27.50 μm for no. 8 to 59.04 μm for no. 12 (Table 4). The longest mean equatorial axis length (E) among the 18 samples belonged to genotype no. 9, and the shortest mean value of E was seen in no. 18 (Table 4). The pollen grain of all the oil-tea genotypes was of medium size, based on the P and E values. The values of P/E ranged from 0.82 to 1.96, and the values of P×E ranged from 900.09 μm$^2$ to 2061.68 μm$^2$ (Table 5). We observed that all the genotypes displayed a triangular or subcircular shape in the polar view, while the equatorial view presented an oblate, oblong, and spheroid shape based on the P/E values (Table 5; Figures 2–4).

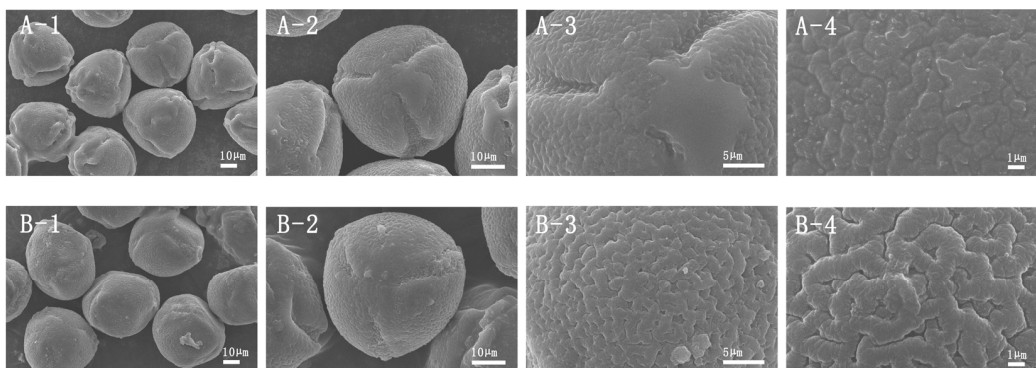

**Figure 2.** *Cont.*

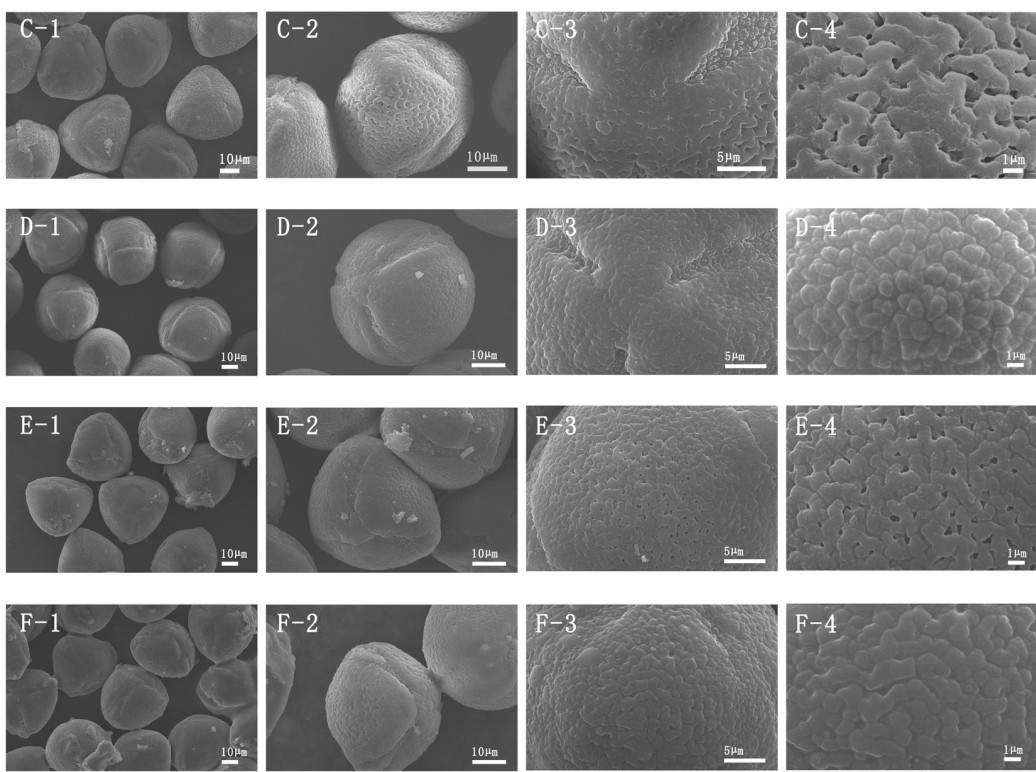

**Figure 2.** Observation of pollen morphology of 18 oil-tea genotypes using a scanning electron microscope, enlarged at ×1000 (first column), ×2000 (second column), ×5000 (third column), and ×10,000 (fourth column). (**A**–**F**) (**1**–**4**), respectively, represent (**A**) (**1**–**4**) *C. gauchowensis* 'HM19', (**B**) (**1**–**4**) is for *C. gauchowensis* 'HM349', (**C**) (**1**–**4**) is for *C. oleifera* 'ASX09', (**D**) (**1**–**4**) is for *C. hainanica* 'PX-6', (**E**) (**1**–**4**) is for *C. meiocarpa* 'ZX0907', and (**F**) (**1**–**4**) is for *C. oleifera* 'DY2'.

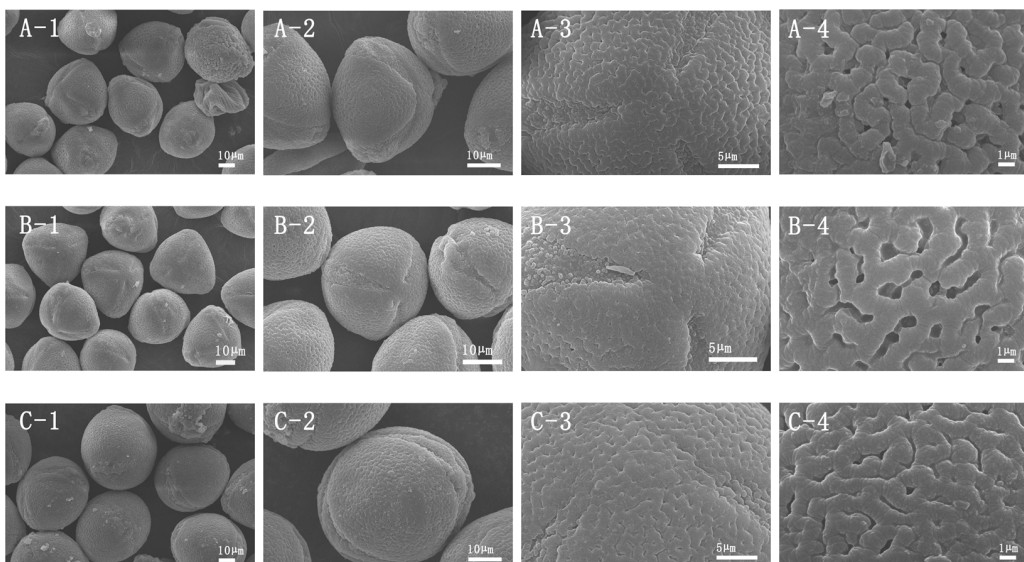

**Figure 3.** *Cont.*

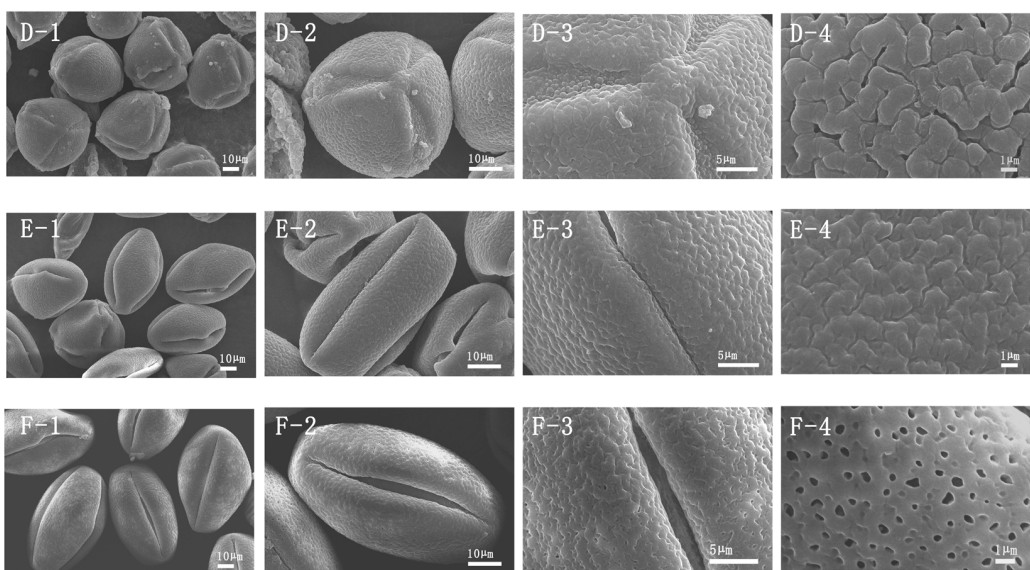

**Figure 3.** Observation of pollen morphology of 18 oil-tea genotypes under different magnifications using a scanning electron microscope, and enlarged at ×1000, ×2000, ×5000, and ×10,000 from left to right, respectively. (**A**–**F**) (**1**–**4**), respectively, represent (**A**) (**1**–**4**) *C. oleifera* 'DZ1H', (**B**) (**1**–**4**) *C. oleifera* 'CY67', (**C**) (**1**–**4**) *C. gauchowensis* 'XW', (**D**) (**1**–**4**) *C. hainanica*, (**E**) (**1**–**4**) *C. magniflora* Chang, and (**F**) (**1**–**4**) *C. mairei* (H. Lév.) Melch. var. lapidea (Y.C. Wu) Sealy.

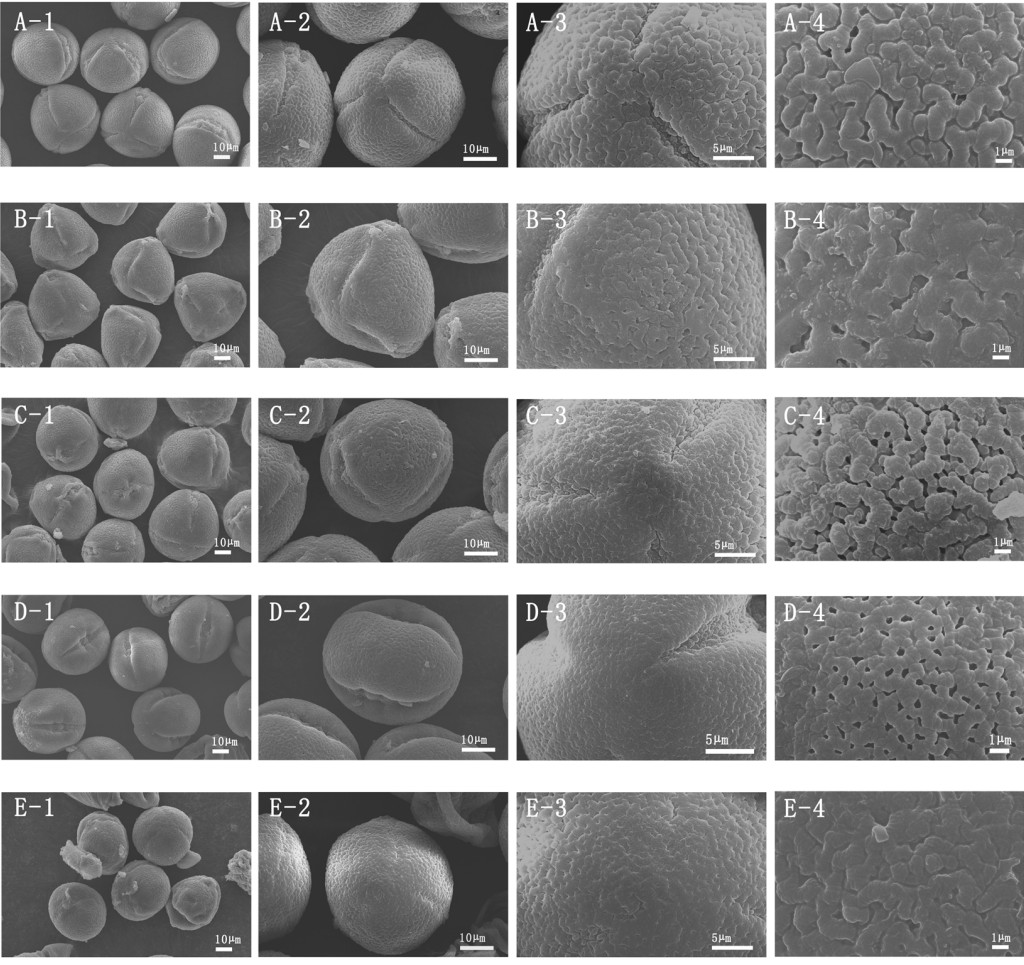

**Figure 4.** *Cont.*

**Figure 4.** Observation of pollen morphology of 18 oil-tea genotypes under different magnifications using a scanning electron microscope, and enlarged at ×1000, ×2000, ×5000, and ×10,000 fold from left to right, respectively. (**A**–**F**) (**1**–**4**), respectively, represent, (**A**) (**1**–**4**) *C. oleifera* 'HJ', (**B**) (**1**–**4**) *C. oleifera* 'HS', (**C**) (**1**–**4**) *C. oleifera* 'HX', (**D**) (**1**–**4**) *C. osmantha*, (**E**) (**1**–**4**) *C. meiocarpa* 'LP', and (**F**) (**1**–**4**) *C. yuhsienensis* Hu.

**Table 4.** Pollen morphology characteristic indices (mean ± SD) (n = 50) of selected oil-tea genotypes.

| Code | Genotype | Polar Axis Diameter (μm) (P) | Equatorial Axis Diameter (μm) (E) | P × E (μm²) |
|---|---|---|---|---|
| no. 1 | *C. gauchowensis* 'HM19' | 31.32 ± 0.34 f | 36.03 ± 0.35 b | 1128.46 ± 11.9 f |
| no. 2 | *C. gauchowensis* 'HM349' | 31.92 ± 0.43 fg | 38.64 ± 0.3 bc | 1233.39 ± 16.1 d |
| no. 3 | *C. oleifera* 'ASX09' | 31.92 ± 0.39 fg | 36.34 ± 0.26 defg | 1273.16 ± 22.4 d |
| no. 4 | *C. hainanica* 'PX-6' | 31.94 ± 0.36 fg | 37.75 ± 0.3 bcd | 1205.74 ± 11.47 e |
| no. 5 | *C. meiocarpa* 'ZX0907' | 30.21 ± 0.44 gh | 35.76 ± 0.51 efgh | 1080.31 ± 14.85 g |
| no. 6 | *C. oleifera* 'DY2' | 28.30 ± 0.34 hi | 34.77 ± 0.4 ghij | 983.99 ± 13.94 h |
| no. 7 | *C. oleifera* 'DZ1H' | 27.98 ± 0.4 i | 33.51 ± 0.4 ijk | 937.61 ± 11.7 h |
| no. 8 | *C. oleifera* 'CY67' | 27.5 ± 0.35 i | 32.76 ± 0.28 k | 900.90 ± 17.22 h |
| no. 9 | *C. gauchowensis* 'XW' | 34.88 ± 0.37 e | 41.62 ± 0.31 a | 1451.71 ± 11.9 c |
| no. 10 | *C. hainanica* | 32.36 ± 0.3 f | 38.74 ± 0.36 b | 1253.63 ± 13.6 d |
| no. 11 | *C. magniflora* Chang | 46.24 ± 1.52 b | 34.47 ± 0.92 hij | 1593.89 ± 32.9 b |
| no. 12 | *C. mairei* (H. Lév.) Melch. var. Lapidea (Y.C. Wu) Sealy | 59.04 ± 0.74 a | 34.92 ± 0.54 fghi | 2061.68 ± 40.14 a |
| no. 13 | *C. oleifera* 'HJ' | 32.73 ± 0.47 f | 37.04 ± 0.47 cde | 1212.32 ± 9.8 de |
| no. 14 | *C. oleifera* 'HS' | 31.01 ± 0.45 fg | 36.75 ± 0.33 de | 1139.62 ± 39.9 ef |
| no. 15 | *C. oleifera* 'HX' | 31.38 ± 0.41 fg | 35.99 ± 0.34 efgh | 1129.37 ± 22.9 f |
| no. 16 | *C. osmantha* | 38.81 ± 0.39 d | 33.14 ± 0.3 jk | 1286.16 ± 10.8 d |
| no. 17 | *C. meiocarpa* 'LP' | 32.49 ± 0.59 f | 36.42 ± 0.75 def | 1183.30 ± 11.8 de |
| no. 18 | *C. yuhsienensis* Hu | 43.17 ± 0.41 c | 21.32 ± 0.42 l | 920.38 ± 44.25 h |

Different letters within a column indicate significant differences at $p \leq 0.05$ via Duncan's multiple range test. Abbreviations: P = polar axis diameter; E = equatorial axis diameter.

**Table 5.** Pollen morphology characteristic indices (mean ± SD) (n = 50) and description of pollen shape of studied oil-tea genotypes.

| Code | Genotype | P/E | Pollen Shape | Polar View | Equatorial View | Exine Surface |
|---|---|---|---|---|---|---|
| no. 1 | *C. gauchowensis* 'HM19' | 0.84 ± 0.01 e | Oblate spheroid | Triangular shape | Oblate shape | Perforate |
| no. 2 | *C. gauchowensis* 'HM349' | 0.83 ± 0.01 e | Oblate spheroid | Subcircular shape | Oblate shape | Perforate |
| no. 3 | *C. oleifera* 'ASX09' | 0.88 ± 0.01 e | Sub-spheroid | Triangular shape | Spherical shape | Perforate |
| no. 4 | *C. hainanica* 'PX-6' | 0.85 ± 0.01 e | Oblate spheroid | Triangular shape | Oblate shape | Verrucate |
| no. 5 | *C. meiocarpa* 'ZX0907' | 0.85 ± 0.01 e | Oblate spheroid | Triangular shape | Oblate shape | Perforate |
| no. 6 | *C. oleifera* 'DY2' | 0.82 ± 0.01 e | Oblate spheroid | Triangular shape | Oblate shape | Verrucate |
| no. 7 | *C. oleifera* 'DZ1H' | 0.84 ± 0.01 e | Oblate spheroid | Triangular shape | Oblate shape | Perforate |
| no. 8 | *C. oleifera* 'CY67' | 0.84 ± 0.01 e | Oblate spheroid | Triangular shape | Oblate shape | Perforate |
| no. 9 | *C. gauchowensis* 'XW' | 0.84 ± 0.01 e | Oblate spheroid | Triangular shape | Oblate shape | Perforate |
| no. 10 | *C. hainanica* | 0.84 ± 0.01 e | Oblate spheroid | Triangular shape | Oblate shape | Perforate |
| no. 11 | *C. magniflora* Chang | 1.39 ± 0.06 c | prolate ellipsoid | Triangular shape | Oblong shape | Perforate |
| no. 12 | *C. mairei* (H. Lév.) Melch. var. Lapidea (Y.C. Wu) Sealy | 1.70 ± 0.04 b | prolate ellipsoid | Subcircular shape | Oblong shape | Perforate |
| no. 13 | *C. oleifera* 'HJ' | 0.89 ± 0.01 e | Sub-spheroid | Subcircular shape | Spherical shape | Perforate |
| no. 14 | *C. oleifera* 'HS' | 0.89 ± 0.01 e | Sub-spheroid | Triangular shape | Spherical shape | Perforate |
| no. 15 | *C. oleifera* 'HX' | 0.88 ± 0.01 e | Sub-spheroid | Subcircular shape | Spherical shape | Perforate |
| no. 16 | *C. osmantha* | 1.10 ± 0.02 d | Sub-spheroid | Subcircular shape | Spherical shape | Perforate |
| no. 17 | *C. meiocarpa* 'LP' | 0.90 ± 0.01 e | Sub-spheroid | Subcircular shape | Spherical shape | Perforate |
| no. 18 | *C. yuhsienensis* Hu | 1.96 ± 0.04 a | prolate ellipsoid | Subcircular shape | Oblong shape | Reticulate |

Different letters within a column indicate significant differences at $p \leq 0.05$ via Duncan's multiple range test. Abbreviations: P/E = polar axis/equatorial axis value.

**Table 6.** Pollen exine sculpture characteristic indices (mean ± SD) (n = 50) of selected oil-tea genotypes.

| Code | Genotype | Perforation Lumina Diameter (µm) (D) | Paraporal Muri Width (µm) (W) | D/W |
|------|----------|--------------------------------------|-------------------------------|-----|
| no. 1 | *C. gauchowensis* 'HM19' | 0.34 ± 0.03 ghi | 1.06 ± 0.04 cd | 0.31 ± 0.01 g |
| no. 2 | *C. gauchowensis* 'HM349' | 0.31 ± 0.01 hi | 1.40 ± 0.12 a | 0.25 ± 0.02 g |
| no. 3 | *C. Oleifera* 'ASX09' | 0.35 ± 0.02 gh | 0.93 ± 0.03 de | 0.37 ± 0.03 de |
| no. 4 | *C. hainanica* 'PX-6' | 0.42 ± 0.02 cdef | 1.05 ± 0.06 cd | 0.40 ± 0.01 de |
| no. 5 | *C. meiocarpa* 'ZX0907' | 0.31 ± 0.02 hi | 0.92 ± 0.03 def | 0.33 ± 0.01 g |
| no. 6 | *C. oleifera* 'DY2' | 0.56 ± 0.03 b | 0.91 ± 0.03 def | 0.62 ± 0.01 bc |
| no. 7 | *C. oleifera* 'DZ1H' | 0.43 ± 0.02 defgh | 1.25 ± 0.06 ab | 0.34 ± 0.01 fg |
| no. 8 | *C. oleifera* 'CY67' | 0.55 ± 0.03 bcd | 1.25 ± 0.03 ab | 0.43 ± 0.01 ef |
| no. 9 | *C. gauchowensis* 'XW' | 0.51 ± 0.04 bcde | 1.05 ± 0.05 cd | 0.49 ± 0.03 de |
| no. 10 | *C. hainanica* | 0.35 ± 0.01 fghi | 1.15 ± 0.05 bc | 0.31 ± 0.01 g |
| no. 11 | *C. magniflora* Chang | 0.29 ± 0.02 i | 1.01 ± 0.05 cde | 0.29 ± 0.01 g |
| no. 12 | *C. mairei* (H. Lév.) Melch. var. lapidea (Y.C. Wu) Sealy | 0.46 ± 0.02 cdefg | 0.86 ± 0.03 ef | 0.53 ± 0.01 cd |
| no. 13 | *C. oleifera* 'HJ' | 0.41 ± 0.02 efghi | 0.95 ± 0.04 def | 0.43 ± 0.01 ef |
| no. 14 | *C. oleifera* 'HS' | 0.55 ± 0.04 bc | 1.29 ± 0.06 ab | 0.42 ± 0.01 ef |
| no. 15 | *C. oleifera* 'HX' | 0.40 ± 0.02 efghi | 0.90 ± 0.04 def | 0.45 ± 0.01 de |
| no. 16 | *C. osmantha* | 0.53 ± 0.02 bcde | 0.77 ± 0.04 f | 0.70 ± 0.01 b |
| no. 17 | *C. meiocarpa* 'LP' | 0.48 ± 0.02 bcdef | 1.02 ± 0.05 cde | 0.47 ± 0.01 de |
| no. 18 | *C. yuhsienensis* Hu | 1.22 ± 0.11 a | 0.90 ± 0.03 def | 1.30 ± 0.09 a |

Different letters within a column indicate significant differences at *p* ≤ 0.05 via Duncan's multiple range test. Abbreviations: D = pollen perforation diameter; W = paraporal muri width; D/W = pollen perforation diameter/paraporal muri width.

**Table 7.** Pollen exine sculpture characteristic indices (mean ± SD) (n = 50) of studied oil-tea genotypes.

| Code | Genotype | Arc Width of Exine (µm) (a) | Arc Height of Exine (µm) (b) | b/a | Germination Furrow Width (µm) (WG) |
|------|----------|----------------------------|------------------------------|-----|-------------------------------------|
| no. 1 | *C. gauchowensis* 'HM19' | 40.94 ± 0.52 b | 13.37 ± 0.43 a | 0.33 ± 0.01 a | 8.88 ± 0.16 a |
| no. 2 | *C. Gauchowensis* 'HM349' | 39.64 ± 0.29 bc | 11.16 ± 0.18 bc | 0.28 ± 0.01 b | 5.76 ± 0.13 f |
| no. 3 | *C. Oleifera* 'ASX09' | 36.56 ± 0.36 efg | 9.93 ± 0.38 de | 0.27 ± 0.01 bcd | 6.87 ± 0.17 cd |
| no. 4 | *C. hainanica* 'PX-6' | 35.62 ± 0.27 gh | 11.52 ± 0.22 b | 0.32 ± 0.01 a | 7.14 ± 0.17 c |
| no. 5 | *C. meiocarpa* 'ZX0907' | 36.98 ± 0.37 ef | 9.29 ± 0.17 efg | 0.25 ± 0.01 cd | 8.57 ± 0.27 ab |
| no. 6 | *C. oleifera* 'DY2' | 37.75 ± 0.39 de | 9.74 ± 0.26 def | 0.26 ± 0.01 bcd | 8.05 ± 0.18 b |
| no. 7 | *C. oleifera* 'DZ1H' | 35.73 ± 0.35 fgh | 8.94 ± 0.29 fg | 0.25 ± 0.01 cde | 8.49 ± 0.18 ab |
| no. 8 | *C. oleifera* 'CY67' | 35.75 ± 0.17 fgh | 8.76 ± 0.26 fgh | 0.24 ± 0.01 cde | 6.20 ± 0.18 ef |
| no. 9 | *C. gauchowensis* 'XW' | 43.10 ± 0.41 a | 11.77 ± 0.18 b | 0.27 ± 0.01 bc | 8.15 ± 0.21 b |
| no. 10 | *C. hainanica* | 40.93 ± 0.32 b | 10.35 ± 0.28 cd | 0.25 ± 0.01 bcd | 6.84 ± 0.27 cde |
| no. 11 | *C. magniflora* Chang | 26.79 ± 0.27 j | 6.81 ± 0.41 i | 0.25 ± 0.01 bcd | 7.25 ± 0.24 c |
| no. 12 | *C. mairei* (H. Lév.) Melch. var. lapidea (Y.C. Wu) Sealy | 27.58 ± 0.36 j | 6.80 ± 0.21 i | 0.25 ± 0.01 cde | 3.02 ± 0.12 g |
| no. 13 | *C. oleifera* 'HJ' | 35.54 ± 0.26 gh | 8.59 ± 0.28 gh | 0.24 ± 0.01 de | 6.12 ± 0.16 f |
| no. 14 | *C. oleifera* 'HS' | 38.64 ± 0.27 cd | 8.51 ± 0.26 gh | 0.22 ± 0.01 e | 6.34 ± 0.14 def |
| no. 15 | *C. oleifera* 'HX' | 34.67 ± 0.34 hi | 8.50 ± 0.25 gh | 0.25 ± 0.01 cde | 6.22 ± 0.12 def |
| no. 16 | *C. osmantha* | 23.51 ± 0.25 k | 7.93 ± 0.26 h | 0.34 ± 0.01 a | 5.81 ± 0.09 f |
| no. 17 | *C. meiocarpa* 'LP' | 33.97 ± 0.68 i | 9.09 ± 0.26 efg | 0.27 ± 0.01 bcd | 6.41 ± 0.17 def |
| no. 18 | *C. yuhsienensis* Hu | 16.57 ± 0.25 l | 4.51 ± 0.11 j | 0.27 ± 0.01 bc | 2.55 ± 0.2 g |

Different letters within a column indicate significant differences at *p* ≤ 0.05 via Duncan's multiple range test. Abbreviations: a = arc width of exine; b = arc height of exine; b/a = curved degree of exine; WG = germination furrow width.

The pollen exine sculpture of the 18 oil-tea genotypes was mainly perforate, except for no. 4, no. 6, and no. 18 (Table 4). The pollen perforation diameter (D) ranged from 0.29 µm to 1.22 µm, and the width of the paraporal muri (W) ranged from 0.77 µm to 1.40 µm (Table 6). Genotypes no. 18 and no. 11 had the maximum D and minimum D/W, respectively. The highest W belonged to genotype no. 14, and the lowest to no. 16 (Table 6). The arc width of the exine (a) ranged from 26.79 µm to 43.10 µm, and the arc height of

the exine (b) ranged from 6.80 μm to 13.37 μm. The value of b/a ranged from 0.22 to 0.34, while WG ranged from 2.55 μm to 8.88 μm (Table 7).

### 3.3. PCA and Cluster Analysis

Differences among the tested samples, based on 11 parameter measurements of their pollen grains and floral organs, were verified using principal component analysis (PCA) and Q-type cluster analysis. The PCA results indicated that the sum of components 1 and 2 accounted for 73.7% of the total variance (Figure 5). The cluster analysis, with a distance (L) of 7, divided these samples into three groups (Figure 6). Group I consisted of 15 genotypes, namely, no. 1–10, and 13–17, Group II consisted of no. 11 and no. 12, while Group III had only no. 18 (Figure 5).

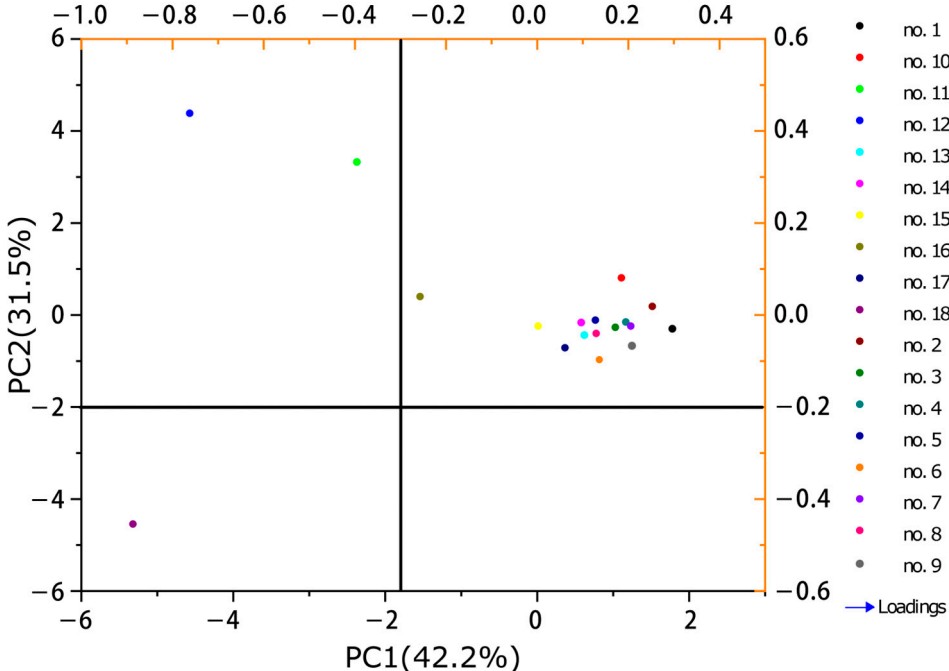

**Figure 5.** Plot of the first two principal component axes from the principal component analysis of floral and pollen characteristics of 18 oil-tea genotypes. Morphological codes correspond to those listed in Table 1.

We analyzed the correlation between the flower morphological characteristics and pollen morphological characteristics of 18 oil-tea genotypes. As shown in Figure 7, there were significant correlations between the flower morphology or pollen exine sculpture and pollen size. The Pearson correlation coefficients (p) between P × E (representing the pollen size) and AN, AH, PH, and SH (representing the flower morphology) were found to be 0.636, 0.781, 0.831, and 0.836 (Figure 7), respectively. Also, the *p*-values between P (representing the pollen size) and a, b, and WG (representing the pollen exine sculpture) were calculated as 0.820, 0.736, and 0.557 (Figure 7), respectively. However, there was a significant negative correlation between p and D with a *p*-value of −0.811 (Figure 7). In addition, we analyzed the correlation between the ploidy and floral organ or pollen characteristics and yet found that the correlation between them was not obvious (Tables 8 and 9).

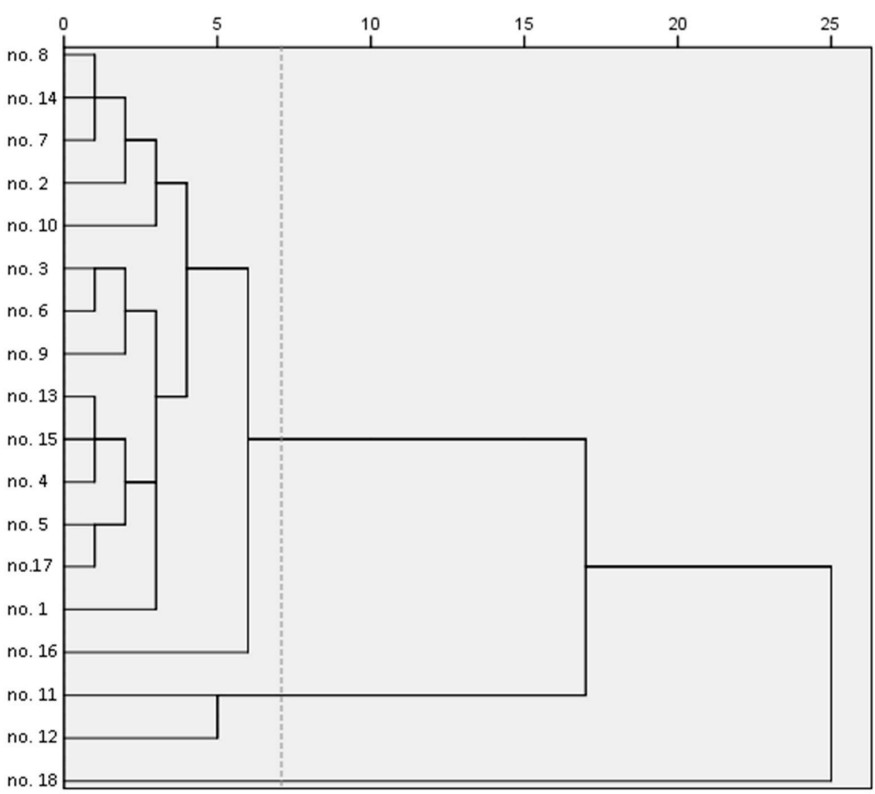

**Figure 6.** Clustering of the 18 oil-tea genotypes based on floral and pollen characteristics is presented in Tables 2–7. Morphological codes *correspond* to these genotype codes listed in Table 1. The dashed line indicates that the Euclidean distance (L) = 7.

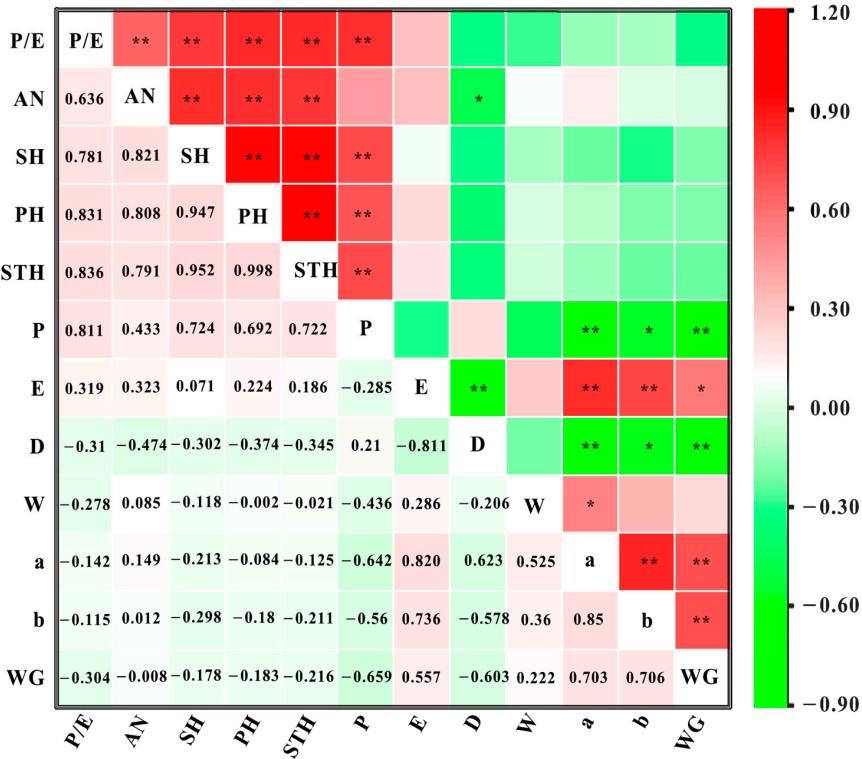

**Figure 7.** Pearson correlation test between floral organs and pollen size or morphology in 18 oil-tea genotypes. * Represents the correlation reached extremely significant ($p < 0.05$). ** Represents the correlation reached extremely significant ($p < 0.01$).

**Table 8.** Pearson correlation test between ploidy and floral organs in selected oil-tea genotypes.

| | | Ploidy | CD | AN | SH | PH | STH |
|---|---|---|---|---|---|---|---|
| Ploidy | Pearson's correlation coefficient (r) | 1 | 0.447 | 0.181 | 0.053 | 0.026 | 0.017 |
| | Significance (p) | 0.000 | 0.063 | 0.473 | 0.834 | 0.919 | 0.948 |
| | Sample number (N) | 18 | 18 | 18 | 18 | 18 | 18 |

Abbreviations: CD = corolla diameter; AN = anther number; SH = stamen height; PH = pistil height; STH = style height.

**Table 9.** Pearson correlation test between ploidy and pollen morphology in selected oil-tea genotypes.

| | | Ploidy | P | E | D | W | a | b | WG |
|---|---|---|---|---|---|---|---|---|---|
| Ploidy | Pearson's correlation coefficient (r) | 1 | −0.136 | 0.271 | −0.052 | 0.073 | 0.334 | 0.414 | 0.302 |
| | Significance (p) | 0.000 | 0.591 | 0.276 | 0.839 | 0.775 | 0.176 | 0.088 | 0.224 |
| | Sample number (N) | 18 | 18 | 18 | 18 | 18 | 18 | 18 | 18 |

Abbreviations: P = polar axis diameter; E = equatorial axis diameter; D = pollen perforation diameter; W = paraporal muri width; a = arc width of exine; b = arc height of exine; WG = germination furrow width.

## 4. Discussion

Floral morphology is very important in the taxonomy and classification of plants [26,27]. In *Malus*, Zhou et al. [26] divided five species into two classes based on flower characteristics. Zhou et al. [28] found that the floral organ size was more representative of the *Malus* classification compared to the shape or number of traits of the flower. In our study, no. 11 and no. 12 can be easily distinguished among the 18 genotypes by their flower color (Figure 1; Table 2). According to the criteria set out by the International *Camellia* Society, the other 16 oil-tea genotypes were classified into three classes based on their corolla diameters (CD), with group I containing no. 5 and 17, group II containing no.1, 2, 4, 13, 14, 15, and 18, while group III had no. 3, 6, 7, 8, 9, 10, and 16 (Table 2). In these groups, we also combined the CD with other floral traits to achieve a more detailed and specific classification for each group. For example, no. 5 and 17 from group I can be further distinguished from each other based on their anther number (AN), stamen height (SH), petal count, and stigma cracking, with no. 5 having more AN, SH, and petal count than no. 17 (Tables 2 and 3). Similarly, in group II, the SN, PH (pistil height), and STH (style height) of flower organs could mark out no. 18, and no. 14 was distinguished by stigma cracking (Tables 2 and 3). In group III, no. 3 and no. 10 were separated from other genotypes by stigma cracking and anther number, respectively (Tables 2 and 3). Given that the other genotypes in groups II and III are not significantly different in floral morphology, we performed a micro-structure analysis for a more detailed classification.

Pollen morphology characteristics played a crucial role in the identification and classification and served as an additional tool for the systematic study of plant groups [24,29,30]. Lubna et al. [16] classified *Vitaceae* based on the palyno-morphological features, including the pollen size, polar and equatorial diameters, pollen shape, and exine sculpturing. In *Iris barbata*, forty-eight cultivars were divided into five types according to the germination furrows [10]. Zhao et al. [31] pointed out that the pollen of oil-tea was of medium size by analyzing the values of P and E. In our study, the pollen size of oil-tea was of medium size based on the P/E values, with the largest size pollen in no. 12 and the smallest size pollen in no. 8. The pollen shape could be reflected by the polar axis length (P), equatorial axis length (E), and polar axis/equatorial axis value (P/E) [9,12,32]. According to the pollen shape, eighteen oil-tea genotypes had three types in the equatorial view: oblong, spherical, and oblate, and had two types in the polar view: a triangular and subcircular shape (Table 5), which is consistent with the results of Lin et al. [33] and Chen et al. [34].

In prior research, the unique characteristics of pollen's exine sculpture were considered a reliable basis in the classification of plant species [35]. According to Fan et al. [36] and Song et al. [12], the exine ornamentation of *C. oleifera* was mostly perforate and occasionally verrucate or microreticulate. The trait of the exine sculpture is commonly described by

the lumina diameter (D), lumina muri width (W), and the ratio of lumina diameter to the muri width (D/W) [9,37,38]. In our report, no. 4 and 6 displayed small verrucations on the exine surface, while the other 16 genotypes were perforate, which had an uneven distribution of perforations in the exine surface, consistent with studies of Liang et al. [33] and Chen et al. [39]. These classes, distinct by their exine surface, were further identified among each other based on the D, W, a (arc width of exine), b (arc height of exine), and WG (germination furrow width) (Table 7). Therefore, in our study, the pollen exine sculpture can be a relevant parameter and beneficial to distinguish various oil-tea genotypes. Moreover, we analyzed the correlation between the flower morphological characteristics and pollen morphological characteristics of 18 oil-tea genotypes. There was a significant positive correlation between the quantitative indicators of flower morphology, such as the AN, SH (Stamen height), PH, STH, and pollen size (including P $\times$ E and P) (Figure 7), consisting of the results of Teixido et al. [40] and Hao et al. [41], which found the larger the flower size of the plant, the larger the pollen size. In addition, we found that there was a strong correlation between pollen size (P, E) and pollen exine sculpture (D, W, a, b, and WG) in the pollen morphology of oil-tea (Figure 7). However, there was no significant correlation between the pollen or flower organ size and plant ploidy (Tables 8 and 9); our results were inconsistent with the results of Nico et al. [42] and Landis et al. [43]. We speculated that the possible reason was due to the relatively complex variety of ploidy in *Camellia*.

The classification of the genus oil-tea has been controversial due to the frequent interspecific hybridization, polyploidy, and morphological variations [44]. Many scholars presented different points in the classification of the *Camellia* genus. Qi et al. [7] divided *C. gauchowensis* and *C. hainanica* into a single group and pointed out that *C. oleifera*, *C. meiocarpa*, and *C. osmantha* evolved from *C. drupifera* (the unifying name of *C. gauchowensis* and *C. hainanica*). Chen et al. [45] stated that *C. osmantha* was a new independent species and had a closer relationship to *C. gauchowensis* and *C. hainanica*. Zhao et al. [46] thought that there was a genetic relationship between *C. japonica* and *C. oleifera*. In our research, the cluster analysis and PCA, based on the eleven pollen and floral organ traits, have grouped all *C. gauchowensis*, *C. oleifera*, *C. hainanica*, and *C. meiocarpa* into Cluster I, except for no. 16 (which belongs to *C. osmantha*), which showed a more distant relationship to other species (Figures 5 and 6). No. 18 was grouped into group II, and no. 11 and no. 12 were divided into group III (Figures 5 and 6), with similar results observed by Yan et al. [47] and Zhu et al. [48]. The results of our cluster analysis and PCA provided important taxonomic information, from a palynological and floral morphological perspective, on the phylogenetic relationships among these 18 oil-tea genotypes.

## 5. Conclusions

Among the 18 oil-tea genotypes, our results indicated that there are significant differences in the size and number of flower organs. The pollen morphology, which was researched under SEM, has a high diversity pattern. The 18 oil-tea genotypes were classified into three categories based on the PCA and cluster analysis of floral and pollen characters. This study enriches our knowledge of palynology and provides reliable information for the classification of oil-tea genotypes.

**Author Contributions:** Conceptualization, Q.Y. and F.Z.; methodology, Q.Y. and F.Z.; software, Q.Y. and Z.P.; resources, Q.Y., F.Z. and Y.L.; supervision, F.Z., D.Y. and H.X.; validation, Q.Y., F.Z. and J.M.; visualization, Q.Y.; writing—original draft, Q.Y.; writing—review and editing, Q.Y., F.Z., H.X., D.Y. and J.M. All authors have read and agreed to the published version of the manuscript.

**Funding:** This work was supported by the Special Funds for Construction of Innovative Provinces in Hunan Province (2021NK1007) and the National Natural Science Foundation of China (No. 32271841).

**Data Availability Statement:** The data presented in this study are available on request from the corresponding author. The data are not publicly available due to privacy.

**Conflicts of Interest:** The authors declare no conflicts of interest.

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
