# Peer review of "Pollen and Floral Organ Morphology of 18 Oil-Tea Genotypes and Its Systematic Significance"

_horticulturae, doi:10.3390/horticulturae10050524_

Round 1
Reviewer 1 Report
Comments and Suggestions for Authors
I think pollen features must be described according modern palynological terminology: W. Punt at al. (2007) and/or Halbritter, H. et al. (2018). So, pollen ornamentation will be mainly rugulate or rugulate-perforate, part of patterns will be verrucate, perforate or reticulate. Other characteristics must be partly rename too. Conclusions can be broader and more specific.
This article provides original descriptions and measurements of floral and pollen morphology. However, pollen morphology has only been studied using a scanning electron microscope; measurements using a light microscope are more applicable. Differences were found at the SEM level between the specimens and species studied, but these are not mentioned in either the abstract or the conclusions. And the terminology should be modern. The main aim of this study is not stated. The possible correlations between ploidy and floral morphological and pollen data were not discussed.
Author Response
Dear reviewers:
We would like to thank you for your kind letter and for reviewers’ constructive comments concerning our article (Manuscript ID: forests-2954625). These comments are all valuable and helpful for improving our article. All the authors have seriously discussed about all these comments. According to the reviewers’ comments, we have tried best to modify our manuscript to meet with the requirements of your journal. In this revised version, changes to our manuscript within the document were all highlighted by using colored text. Point-by-point responses to the reviewers are listed below this letter.
Comments1: I think pollen features must be described according modern palynological terminology: W. Punt at al. (2007) and/or Halbritter, H. et al. (2018). So, pollen ornamentation will be mainly rugulate or rugulate-perforate, part of patterns will be verrucate, perforate or reticulate. Other characteristics must be partly rename too. Conclusions can be broader and more specific.
Response1: Thank you for pointing this out. We agreed with this comment. The reference was changed to Halbritter, H. et al. (2018), and these palynological terminologies had been rephrased as: “ The pollen grains of oil-tea are monad, triangular or subcircular in polar view and oblate, spherical or oblong in equatorial view, and of medium grade in pollen size. The pollen exine sculpture is perforate, verrucate, and reticulate.”, and other characteristics in the article has also been modified according to modern palynological terminology. Meanwhile, we also revised conclusions to make it more broader and specific.
Comments2: This article provides original descriptions and measurements of floral and pollen morphology. However, pollen morphology has only been studied using a scanning electron microscope; measurements using a light microscope are more applicable.
Response2: Thank you for your suggestion. Our original intention is to discuss the floral organs and pollen morphology of oil-tea. So we choose to use survey methods to describe gross feature of flower organs and SEM to explore pollen morphology microscopically.
Comments3: Differences were found at the SEM level between the specimens and species studied, but these are not mentioned in either the abstract or the conclusions. And the terminology should be modern. The main aim of this study is not stated. The possible correlations between ploidy and floral morphological and pollen data were not discussed.
Response2: Thank you for your reminder. We added the statement of the main aim and the species studied at the SEM level in the abstract. In addition, we revised the discussion, in which we analyzed the relationship between ploidy and floral morphological and pollen data.
Thank you and best regards.
Yours sincerely,
Qian Yin
20221100088@csuft.edu.cn
Corresponding author:
Feng Zou
t20142217@csuft.edu.cn
Reviewer 2 Report
Comments and Suggestions for Authors
Dear Authors,
Unfortunately the text is very difficult to read. The abbreviation should be removed from the Discussion. It is necessary to explain the abbreviation. And also necessary to unify all term (especially palynological). The results of the suggest that the conclusions can be improved.

Author Response
Dear reviewers:
We would like to thank you for your kind letter and for reviewers’ constructive comments concerning our article (Manuscript ID: forests-2954625). These comments are all valuable and helpful for improving our article. All the authors have seriously discussed about all these comments. According to the reviewers’ comments, we have tried best to modify our manuscript to meet with the requirements of your journal. In this revised version, changes to our manuscript within the document were all highlighted by using colored text. Point-by-point responses to the reviewers are listed below this letter.
Comments1: Unfortunately the text is very difficult to read. The abbreviation should be removed from the Discussion. It is necessary to explain the abbreviation. And also necessary to unify all term (especially palynological).
Response1: Thank you for pointing these out error. These errors had been corrected in the manuscript.
Comments2: The results of the suggest that the conclusions can be improved.
Response2: Thank you for your reminder. We had corrected conclusions. The conclusions had been rephrased as: Among the 18 oil-tea genotypes, our results indicated that there are significant differ-ences in the size and number of flower organs. And pollen morphology, which was researched under a SEM, has a high diversity pattern. These tested 18 oil-tea genotypes were classified into three categories based on the PCA and cluster analysis of floral and pollen characters. This study enriches our knowledge of palynology and provides reli-able information for the classification of oil-tea genotypes.
Thank you and best regards.
Yours sincerely,
Qian Yin
20221100088@csuft.edu.cn
Corresponding author:
Feng Zou
t20142217@csuft.edu.cn
Reviewer 3 Report
Comments and Suggestions for Authors
Dear Authors,
I have carefully read your paper concerning Camellia classification based on floral and pollen charcteristics.
My overall mpression is positive, you conducted laborous experiments which gave a lot of data. The data was elborated and interpret properly. In my opinion, your experiments could have been enriched with genetic studies, eg. based on molecular markers, but since this was not your aim, I find the methods used by you as sufficient.
Nonetheless, this manuscript needs revision, particularly in terms of data presentation, since this can be done more clear for better understanding and interpretation by the reader.
My suggestions are listed below in the order of appearance in the manuscript.
Title: instead of 18 use "selected"
line 63 - Omit "These"
65 - "The ploidy level of the selected specimens (or accesions) were described by...." Explain also, if this is the ploidy level of pollen or somatic tissues
Fig1. The letters on the photo do not correspond with the order of genotypes presented in the further tables. Please, use the same order and labelling of genotypes - either letters or numbers.
Tab.1. The sampling date is not necessary here, ploidy level should be given as 2n=8x, 2n=10x, etc. provided they refer to somatic tissues.
In all the references to tables or figures use "selected" or "studied" instead of "18".
118 - six? what six? I can see only three indices.
Tab. 2 and aal the other tables - Start the reference with the most important information - what is present in the table, eg. "Flower morphology indices (mean +- SD) and floral organ characteristics of studied oil-tea genotypes." Give information about n=20 below the table, where you mention the statistics information. I prefer to avoid the usage of letter abbreviation of column headings, since the reader has to dig under the table to find out what it shows. I suggest to use full name of the trait, eg. "corolla diameter (mm)", "stamen height (mm)", etc. Omit mean+-SD, since it is expalind in the table reference.
For "Petal count per flower" give just basic number +-1 (if it occurs), eg. 6+-1.
My general suggestion is to present the data in the form of garphs, think it over, the garphs are always more informative than tables.
140 - square should be as the upper index digit
Table 4. There is a mess in tersm of types for perticular views and exine surface, they cannot be name the same like "Type 1", it is hard to be read with understanding. In this case I suggest to use abbreviations.
Fig. 6 There is letter C. for "Camellia" missing in the species names.
The literature is sufficient.
I have also one more concern, which is mainly attributed with the ethics of publications. I figured out that one of the author, JM, contributed only in the writing process. In my opinion this kind of contribution is too small according to the whole work - conceptualization, design, performance, data collection, etc to be granted with the authorship. The editing and writing is rather a type of linguistic skill, not necessarily related to the matter of plant science. I will share this doubt with the Editor and let them make a decission on this matter.
Regards,
Comments on the Quality of English LanguageEnglish writing is fine. Two minor flaws are indictaed in the comments for Authors.
Author Response
Dear reviewers:
We would like to thank you for your kind letter and for reviewers’ constructive comments concerning our article (Manuscript ID: forests-2954625). These comments are all valuable and helpful for improving our article. All the authors have seriously discussed about all these comments. According to the reviewers’ comments, we have tried best to modify our manuscript to meet with the requirements of your journal. In this revised version, changes to our manuscript within the document were all highlighted by using colored text. Point-by-point responses to the reviewers are listed below this letter.
Comments1: The results of the suggest that the conclusions can be improved.
Response1: Your suggestions had been very helpful to us. The conclusions had been checked and corrected.
Comments2: The letters on the photo do not correspond with the order of genotypes presented in the further tables. Please, use the same order and labelling of genotypes - either letters or numbers.
Response2: Thank you for pointing out this error. We have modified the position of the photo annotation to make it consistent with the order position of sample of the tables.
Comments3: Tab.1. The sampling date is not necessary here, ploidy level should be given as 2n=8x, 2n=10x, etc. provided they refer to somatic tissues. In all the references to tables or figures use "selected" or "studied" instead of "18". 118 - six? what six? I can see only three indices.(ok)
Response3: Thanks for your suggestion. These errors had been corrected in the manuscript.
Comments4: Tab. 2 and aal the other tables - Start the reference with the most important information - what is present in the table, eg. "Flower morphology indices (mean +- SD) and floral organ characteristics of studied oil-tea genotypes." Give information about n=20 below the table, where you mention the statistics information. I prefer to avoid the usage of letter abbreviation of column headings, since the reader has to dig under the table to find out what it shows. I suggest to use full name of the trait, eg. "corolla diameter (mm)", "stamen height (mm)", etc. Omit mean+-SD, since it is expalind in the table reference. For "Petal count per flower" give just basic number +-1 (if it occurs), eg. 6+-1. My general suggestion is to present the data in the form of garphs, think it over, the garphs are always more informative than tables.
Response4: Thank you for pointing out this detail. We changed all the tables as required by the standard (start the reference with the most important information - what is present in the table).
Comments5: 140 - square should be as the upper index digit Table 4.
Response5: Thank you for pointing out this error. These errors had been corrected in the manuscript.
Comments6: There is a mess in tersm of types for perticular views and exine surface, they cannot be name the same like "Type 1", it is hard to be read with understanding. In this case I suggest to use abbreviations.
Response6: I agree with this suggestion. We have changed all "Type" in manuscripts and replaced "Type" with the corresponding abbreviations.
Comments7: Fig. 6 There is letter C. for "Camellia" missing in the species names.
Response7: Thanks for your suggestion and Figure 6 has been modified.
Thank you and best regards.
Yours sincerely,
Qian Yin
20221100088@csuft.edu.cn
Corresponding author:
Feng Zou
t20142217@csuft.edu.cn